# Emergency Disposal Solution for Control of a Giant Landslide and Dammed Lake in Yangtze River, China

**Guiya Chen** [1,2], **Xiaofeng Zhao** [3], **Yanlai Zhou** [2,4,*], **Shenglian Guo** [2], **Chong-Yu Xu** [4] and **Fi-John Chang** [5]

1   Changjiang Water Resources Commission, Wuhan 430010, China; chengy@cjh.com.cn
2   State Key Laboratory of Water Resources and Hydropower Engineering Science, Wuhan University, Wuhan 430072, China; slguo@whu.edu.cn
3   Hubei Provincial Water Resources and Hydropower Planning Survey and Design Institute, Wuhan 430064, China; zyl23bulls@163.com
4   Department of Geosciences, University of Oslo, P.O. Box 1047 Blindern, N-0316 Oslo, Norway; c.y.xu@geo.uio.no
5   Department of Bioenvironmental Systems Engineering, National Taiwan University, Taipei 10617, Taiwan; changfj@ntu.edu.tw
*   Correspondence: yanlai.zhou@whu.edu.cn

**Abstract:** Although landslide early warning and post-assessment is of great interest for mitigating hazards, emergency disposal solutions for properly handling the landslide and dammed lake within a few hours or days to mitigate flood risk are fundamentally challenging. In this study, we report a general strategy to effectively tackle the dangerous situation created by a giant dammed lake with 770 million cubic meters of water volume and formulate an emergency disposal solution for the 25 million cubic meters of debris, composed of engineering measures of floodgate excavation and non-engineering measures of reservoirs/hydropower stations operation. Such a disposal solution can not only reduce a large-scale flood (10,000-year return period, 0.01%) into a small-scale flood (10-year return period, 10%) but minimize the flood risk as well, guaranteeing no death raised by the giant landslide.

**Keywords:** water resources management; landslide; dammed lake; flood risk

## 1. Introduction

Landslides can be attributed to rainstorms [1–3], earthquakes [4,5], avalanches, human construction activities, land-use change [6–8] as well as natural processes of erosion that ruin land slopes [9,10]. Furthermore, large landslides often block river vales, giving rise to the formation of giant dams, so that catastrophic debris flows and floods would easily take place in case of dam break [11].

Early warning, in close association with theoretical considerations, predicts that the intensity of giant landslides will increase in a climate and anthropogenic changes setting according to physical experiments and post-assessment [12,13]. The potential for a giant landslide intensification with earthquake, rainfall-induced storm runoff, and avalanche is of significant societal concern, with a huge dammed lake inducing dam break, flash floods and debris being one of the costliest and dangerous natural hazards worldwide [14]. Landslides are one of the most difficult natural disasters to predict since the factors that affect slope stability vary dramatically in both space and time. An emergency disposal solution for adequately handling the landslide and dammed lake within a few hours or days to reduce flood risk is fundamentally challenging, despite landslide early warning and post-assessment being of great interest for mitigating hazards. Nevertheless, the expected general strategy in response

to landslides and dammed lake extreme intensification is still unclear. Hence, we propose that more attention should be paid to increase infrastructure resilience to our changing environment, as observations and field surveys of storm runoff extremes, earthquakes and avalanches show that they would cause major challenges for existing infrastructure systems.

In the stream nearby the Baige Village of the Tibet Autonomous Region, which suffers from the obstacles caused by massive landslides, bulldozers are excavating a diversion channel to reduce the risk of flooding due to the fast-rising water levels of the blocked Jinsha River, which is situated in the upper reach of the Yangtze River (Figure 1). On 3 November, 2018, the landslide following the first landslide, which occurred on 11 October, 2018, caused about 25 million cubic meters of debris to hurtle down a mountainside, which created a natural dam that was 58.24 m high, 195 m wide and 273 m long, situated above the elevation of 2966 m along the Jinsha River. "I was surprised that the water level of the lake increased so fast at a speed of 0.7 m per hour" said one of the hydrologists from the Changjiang Water Resources Commission (CWRC) who first glimpsed the dammed lake. In the next few days, the water volume of this dammed lake increased from 300 million cubic meters and reached 770 million cubic meters.

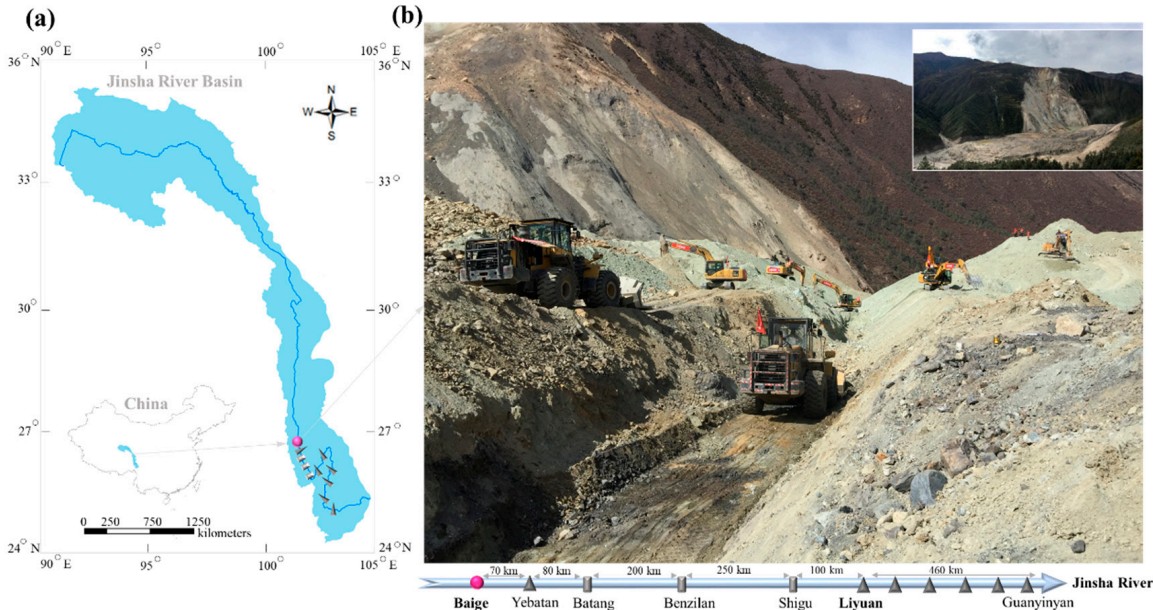

**Figure 1.** Location of landslide and excavation of floodgate. (**a**) Location of landslide; (**b**) workers carved out a diversion channel through the landslide blocking the Jinsha River. Baige is the place where the landslide occurred, and Yebatan is a hydropower station. Batang, Benzilan, and Shigu are hydrological stations. Liyuan and Guanyinyan are the first and the last reservoirs of the six cascade reservoirs in the midstream of the Jinsha River, respectively.

In this study, we propose an emergency disposal solution combining structural and non-structural measures to reduce the flood risk encountered in a giant landslide and dammed lake. The rest of this study is arranged as below. Section 2 introduces the used methods, including the structural measures and the non-structural measures, respectively. Section 3 shows the results on the methods employed to control the landslide and dammed lake while discussing the future challenges of landslide management in the Yangtze River.

## 2. Methods

The goal of this study is to propose an emergency disposal solution for landslide control, including structural and non-structural measures and reducing flood risk to a safe range. Figure 2 illustrates the

emergency disposal solution architecture that integrates the structural measures (Figure 2a) with the non-structural measures (Figure 2b). The used methods are briefly described below.

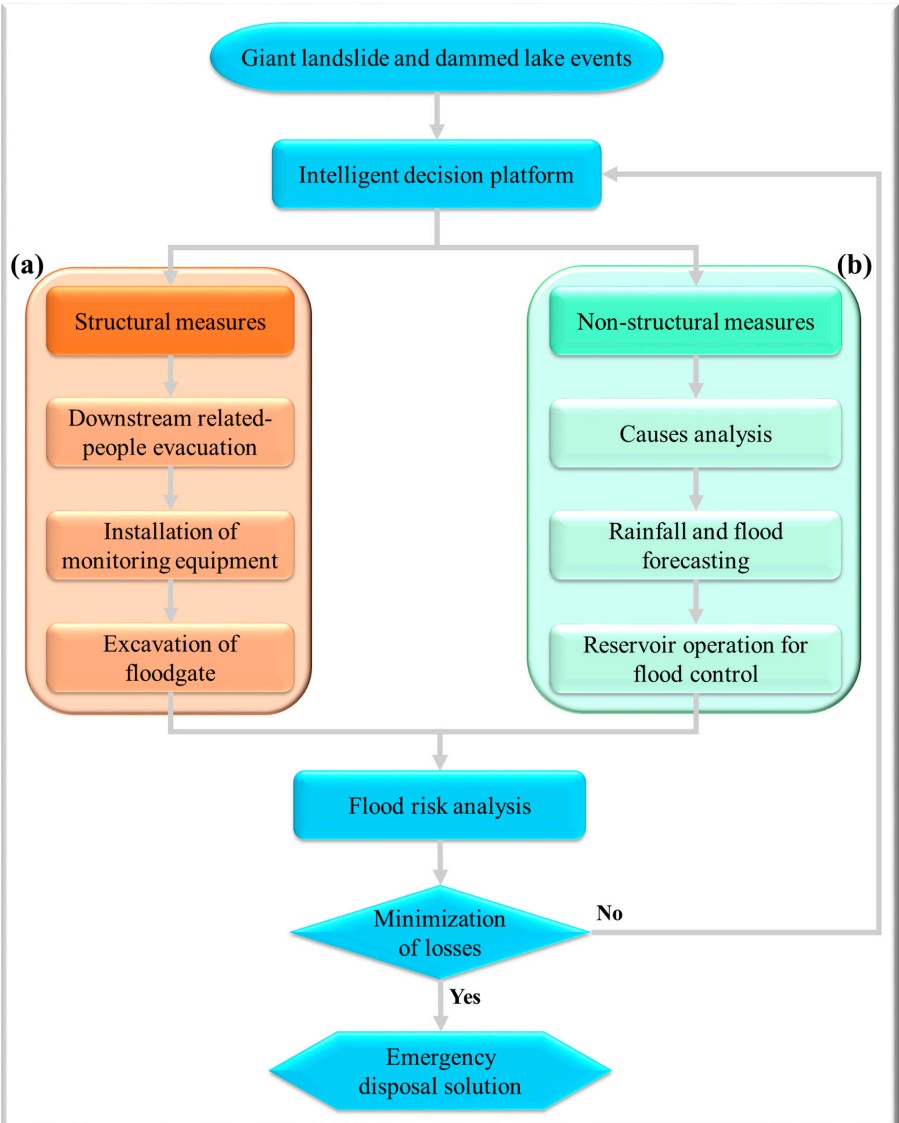

**Figure 2.** Emergency disposal solution for landslide control. (**a**) Structural measures; (**b**) non-structural measures.

*2.1. Structural Measures*

　　The most commonly used structural measures are classified by the sort of slope stabilization methods used, including the geometric method [15,16], chemical and mechanical method [17], and hydrogeological method [18]. The former two methods are preferentially adopted to mitigate potential landslides, while the third method is preferentially adopted to handle breakout landslides [19]. Therefore, in this study, a package of hydrogeological methods is employed to mitigate landslide and dammed lake. The hydrogeological methods consist of downstream related-people evacuation, installation of monitoring equipment, as well as excavation of floodgate. The downstream related-people evacuation can effectively lower life and property losses ahead of natural disasters extension. Mobile hydrological monitoring equipment is installed to collect real-time rainfall and flood information. The excavation of floodgate can be used to rapidly decrease the water volume of the dammed lake after implementing landslide surface protection and control of surface erosion mitigation measures.

## 2.2. Non-Structural Measures

The non-engineering measures consist of analysis of the causes, rainfall and flood forecasting and reservoir operation for flood prevention. Firstly, a clear understanding of the processes that caused the landslide (i.e., cause analysis) plays a vital role in the selection and design of appropriate, cost-effective remedial measures. While the destabilizing processes are grouped into slow changing (e.g., weathering, erosion) and fast-changing processes (e.g., earthquake, drawdown) [20]. Secondly, the early warning system for rainfall and flood forecasting can make it possible to implement a regional real-time assessment of landslide hazard threats [21]. Last but not least, flood prevention and pre-discharge measures for reservoirs (hydropower stations) can be used to decrease the flood risk of downstream area—especially when the breakout landslide has further caused a dammed lake in the river.

After several structural and non-structural measures are formulated and designed to cope with the breakout landslide, it is imperative to undertake a flood risk analysis [22–25] for selecting an appropriate and cost-effective emergency disposal solution from the standpoint of minimizing losses.

## 3. Results and Discussion

### 3.1. Causes Analysis for Giant Landslide

First—considering the earthquake factor—Tibet in China is an earthquake-prone area, and a great number of earthquakes (642, respectively) larger than magnitude 5.0 occurred in Tibet during 1900 and 2017. Out of these earthquakes, there were 503 measuring a magnitude of 5.0–5.9, 130 measuring 6.0–6.9, 7 measuring 7.0–7.9, and two measuring 8.0–8.9 (Figure 3).

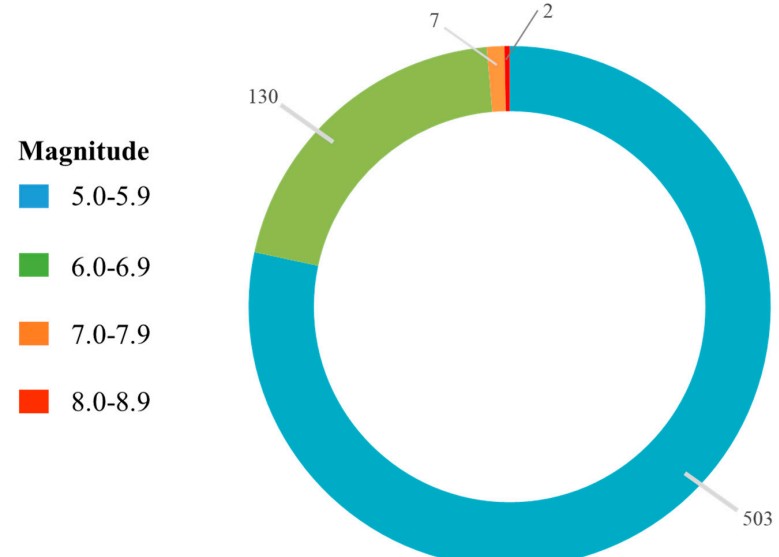

**Figure 3.** Data on earthquakes in Tibet. Data on earthquakes larger than magnitude 5.0 in Tibet during 1900 and 2017, extracted from the United States Geological Survey (USGS) [26] and the China Earthquake Networks Center (CENC) [27].

According to the statistical data released by the USGS, the Chayu County of Tibet underwent a magnitude 8.6 earthquake on 15 August, 1950—which was the most massive earthquake in China to date. Besides, the Naqu City of Tibet was hit by two earthquakes of magnitudes 4.1 and 4.2 on the 6 and 7 October, 2018, respectively. The epicenters of the two earthquakes were 1000 km away from the Baige Village. Second—considering the weather factor—between 11 October and 3 November 2018, some heavy rainfalls (100–250 mm/day) occurred in succession. Therefore, in consequence of the two

factors associated with earthquake and storm precipitation, a landslide struck the Baige Village and initiated the formation of a giant dammed lake.

### 3.2. Emergency Disposal Solution

The director of the Changjiang Water Resources Commission (CWRC) and experts from the Changjiang River Defense General formulated an emergency disposal plan for the dammed lake to properly handle disasters for minimizing losses, as addressed below: (1) organize an evacuation of the affected areas in Tibet, Sichuan, and Yunnan Provinces. (2) Undertake hydrological emergency monitoring, forecasting, and early warning in the upper and lower reaches of the dammed lake. (3) Quickly tackle the dangerous situation raised by the dammed lake and formulate an emergency disposal plan for the stagnation. (4) Implement flood prevention and pre-discharge measures for reservoirs (hydropower stations) that are operational and under construction in the lower reaches of the dammed lake.

On 4 and 5 November, 2018, combining the aerial videos, images and in-situ measurement data collected by drones with the development of the water storage in the dammed lake, it was concluded that without human intervention, the possibility for the dammed lake to collapse would be very high. Besides, the anticipated magnitude of the flood-induced by such a collapse would be enormous, and would bring significant economic losses to the downstream. During 4 and 5 November, 2018, after a comprehensive site-specific survey of the landslide (Figure 4), the director of the CWRC said, "in such emergency case, we only have a few days to assess landslide hazards and figure out what to do".

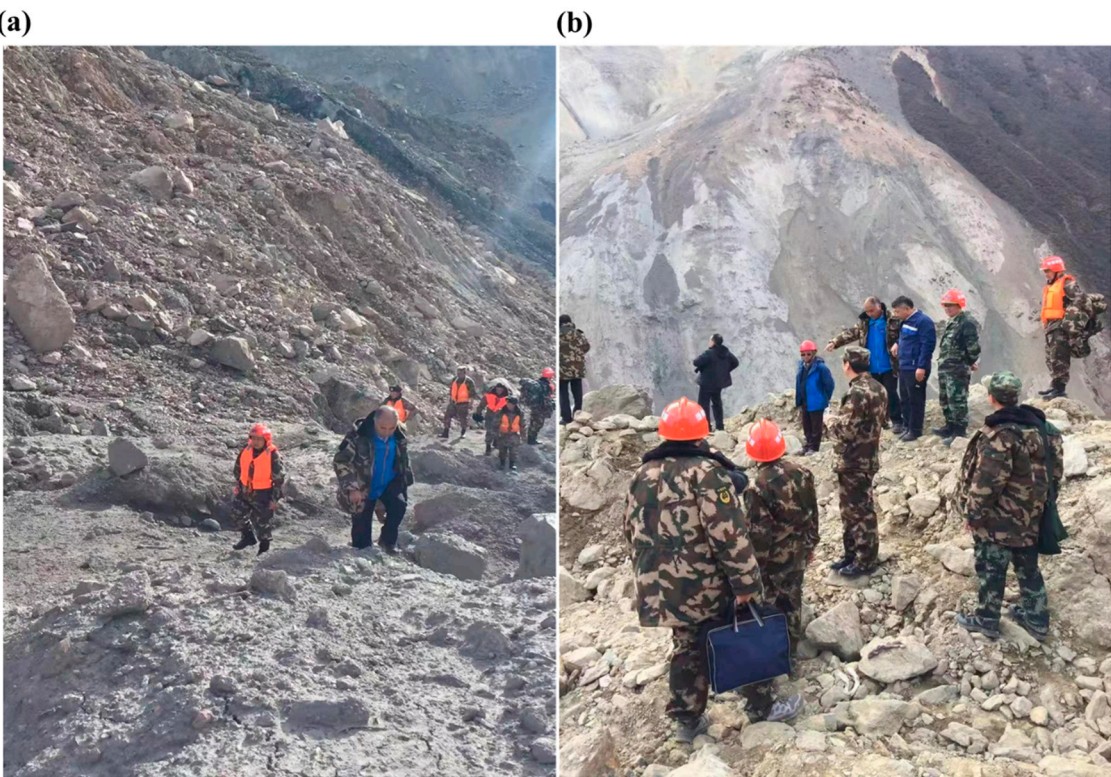

**Figure 4.** Field survey of the Baige landslide. Engineering geologists and hydrologists carried out a field survey of the Baige landslide during (**a**) 4 November, 2018 and (**b**) 5 November, 2018.

The CWRC dispatched 245 people to participate in the disposal of the dammed lake. On 9 November, 2018, three kinds of diversion channels and flood risks were analyzed and formulated, based on three scenarios designed to reduce the lake water level by 5, 10, and 15 m, respectively. Flow peaks of the main control sites along the downstream of the dammed lake were predicted in case of a

dam break as well. On 10 November, 2018, about 80 sets of equipment—including automatic gauges designed for high water level and water gauges—were installed and the forecasts of water level, flow, and possible inundated areas were made (Figure 5). On 11 November, 2018, the engineering team succeeded in digging out a diversion channel with a length of 220 m, the maximum depth of 15 m, a width of 42 m at the top, and a width of 4 m at the base so that floodwater could be released for decreasing the lake water level as well as reducing the flood risks at downstream areas (Supplementary Video S1). Meanwhile, the government quickly evacuated more than 50,000 people downstream of the Baige area and at least 18,657 people in Lijiang City of Yunnan Province.

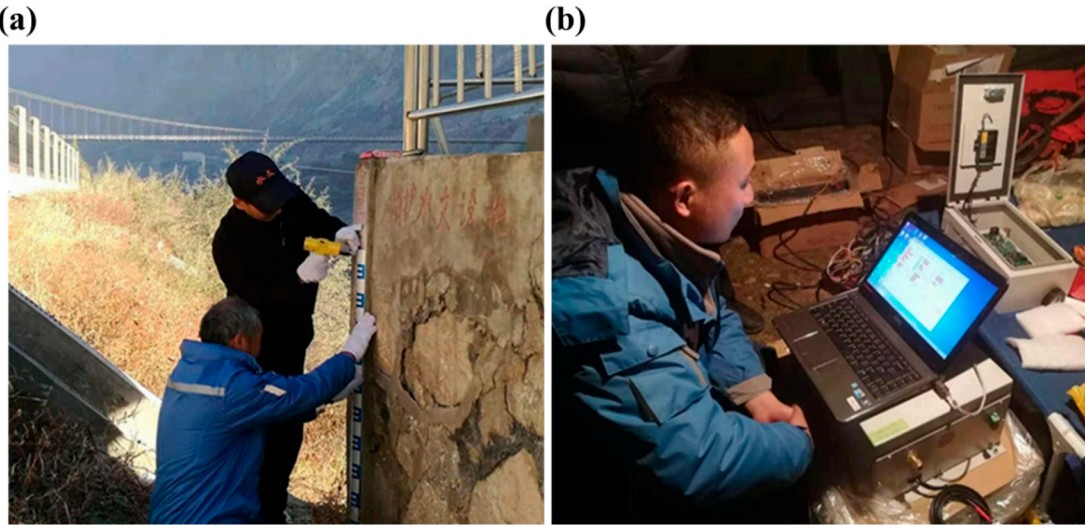

**Figure 5.** Hydrological monitoring in the downstream of Baige landslide. (**a**) Installation of water level ruler; (**b**) testing hydrological monitoring equipment.

Before the dam broke on 13 November, 2018 (Supplementary Video S2), four operational commands on cascade reservoirs were issued by the CWRC to cope with the dam break based on in-situ survey and hydrological forecasting, as described below. To deal with the floods caused by dam break, a series of reservoir operation was implemented started from 5 November, 2018. Three cascade reservoirs (Liyuan, Ahai, and Jinanqiao) vacated a total storage capacity of 0.62 billion cubic meters on 5 November, 2018. It followed six cascade reservoirs (Liyuan, Ahai, Jinanqiao, Longkaikou, Ludila, and Guanyinyan) that vacated a total storage capacity of 1.03 billion cubic meters on 10 November, followed by 1.189 billion cubic meters on 14 November, and finally reached 1.3 billion cubic meters on 15 November.

A few hours after the dam break, the peak flow discharge reached 28,300 cubic meters per second (cms) at the Yebatan hydropower station (19:00, 13 November), 20,900 cms at the Batang station (02:00, 14 November), 15,700 cms at the Benzilan station (13:00, 14 November), 7170 cms at the Shigu station (08:00, 15 November), and 7410 cms in the Liyuan Reservoir (10:00, 16 November), respectively. We noticed that reservoir inflow increased by 30% to 60% at the above-mentioned upstream reservoirs, whereas it only increased 5% to 20% at downstream stations, as compared with historical maximal peak flows. Taking the Liyuan Reservoir as an example, the decreasing rate of the flood peak reached 40%, based on the maximal inflow (7410 cms) and outflow (4490 cms) (Figure 6 and Supplementary Video S3). In other words, the joint operation of the six cascade reservoirs released a total of 1.3 billion cubic meters in storage capacity for flood control would pave the way not only to reduce a large-scale (10,000-year return period) flood into a small-scale (10-year return period) flood but to minimize the flood risk raised by the Baige landslide as well.

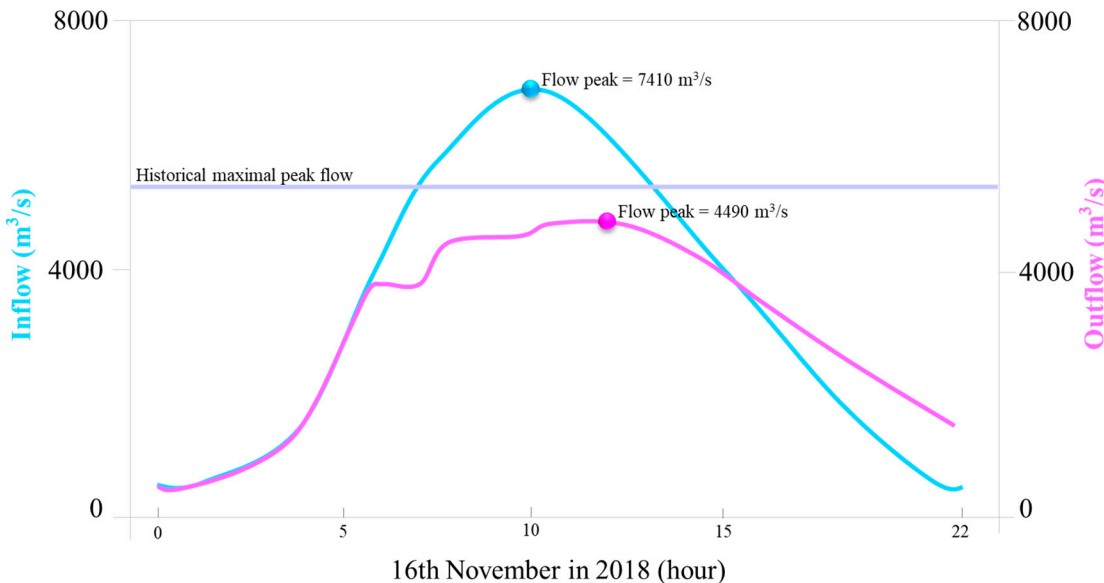

**Figure 6.** Sketch map of reservoir operation for releasing water in the Liyuan Reservoir.

The results indicate that the flood risk and life-property loss are significantly reduced if the structural and non-structural measures are contained and a wide scope of stakeholders are involved in response to the destruction caused by water-related natural catastrophes. The effectiveness of the structural and non-structural measures in such a case is due to a variety of reasons: the downstream related-people evacuation can play a vital role in helping avert losses from flooding and debris; the fast installation of mobile hydrological monitoring equipment can provide first-hand information of real-time hydro-meteorological data for flood forecasting; the excavation of floodgate can effectively address floods resulting from the failure of the dammed lake; the physical cause analysis results can help managers and decision-makers to figure out remedial measures according to the causes and effects of landslides; the early warning system based on hydrological forecasts can make sufficient lead time for diverse sectors and stakeholders to respond if flooding is severe, widespread and sudden; and reservoir operation (e.g., pre-discharge measures) can provide flood control capacity for coping with the floods, in case the dam resulting from the landslide breaks.

In the coming months, the director of the CWRC and his team will assess landslide hazards, reconstruct the damaged monitoring facilities, and revise the safety standards against landslides and dammed lakes. In addition, follow-up studies will pay special attention to predicting rainfall and flood used machine learning techniques [28–30], simulating and/or optimizing reservoirs operation based on advanced intelligent algorithms [31,32], and assessing the impacts of dammed lakes and dam collapse on the hydrological conditions [33,34], sediment deposition and erosion [35,36] as well as biodiversity [37] of the Yangtze River. After these studies were accomplished, the director of the CWRC said, "new achievements will contribute to landslide management in other countries or regions of the world". An integration of landslide-risk-management measures demands the participation of diverse sectors (water management, spatial planning, and emergency management) and diverse governmental levels to offer appropriate policy instruments. Improving cooperation and coordination between policy sectors and administrative levels will benefit to cope with natural hazards. Besides, from the standpoint of methodological transferability, the package of the structural and non-structural measures—apart from the reservoir operation—can be directly applied for the management of other landslide cases or studies, whereas the reservoir operation can only be considered as a special measure against landslide in response to the landslide and a dammed lake raised in the river.

## 4. Conclusions

Sudden landslides with large rock avalanches often block river valleys and result in the formation of large dammed lakes, which would cause devastating debris flows and floods once the dams collapse. The loss of life and destruction of property caused by catastrophic debris flows around the world will most likely continue intensifying as the world population increases, urban development boosts, deforestation expands and land-use alternates. In this case report, we proposed an emergency disposal solution for adequately processing the landslide and dammed lake. The Baige landslide and dammed lake upstream of the Yangtze River of China was selected as a study case. The contributions of the proposed solution consisted of: (1) the emergency disposal solution's ability to integrate the structural and non-structural measures; and (2) that the emergency disposal solution can significantly reduce the flood risk and property losses caused by the landslide and dammed lake.

Despite the Baige landslide created a giant dammed lake with 770 million cubic meters of water volume, the formulated emergency disposal solution can adequately remove the 25 million cubic meters of debris and effectively release the water volume. Furthermore, the hybrid of engineering measures of floodgate excavation and non-engineering measures of reservoirs/hydropower stations operation can decrease a large-scale flood (10,000-year return period, 0.01%) into a small-scale flood (10-year return period, 10%) to minimize the flood risk.

Within only two weeks, China succeeded in coping with the life-threatening event of the giant landslide-induced dammed lake that took place in the Baige Village along the Yangtze River, guaranteeing that no death was caused by the landslide. Besides, this provided valuable experiences and effective strategies to minimize disaster damages, and the losses of life and property resulted from giant landslides stimulated by climate and anthropogenic changes.

**Supplementary Materials:** The following are available online at http://www.mdpi.com/2073-4441/11/9/1939/s1, Video S1: Succeeded in excavating the diversion channel. After working round the clock for one week, workers succeeded in excavating the diversion channel through the Baige Dam on 11 November 2018. Video S2: Break of the Baige dam. After the human intervention, the Baige dam started to break on 13 November 2018. Video S3: A swollen Jinsha River gush through the Liyuan reservoir. In response to the Baige dam break, the Liyuan Reservoir started to release floodwaters in advance on 13 November 2018. Data and materials availability: The data of global earthquakes can be downloaded from the United States Geological Survey (USGS, https://earthquake.usgs.gov) and the China Earthquake Networks Center (CENC, http://www.cenc.ac.cn, Chinese). All data needed to evaluate the results and conclusions in the manuscript are provided in the manuscript or the Supplementary Materials.

**Author Contributions:** G.C., X.Z. and Y.Z. carried out the analysis and wrote the article, S.G., C.X. and F.C. provided technical assistance and contributed in writing the article, G.C. carried out the field survey.

**Funding:** This research was funded by the National Key Research and Development Program of China (2018YFC0407904), the Research Council of Norway (FRINATEK Project 274310) and the Excellent Youth Science Foundation of NSFC (Number 51822908).

**Acknowledgments:** We thank the Changjiang Water Resources Commission (CWRC) of China for providing the monitoring data and video materials of field survey.

**Conflicts of Interest:** The authors declare no conflict of interest.

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
