# Peer review of "Emergency Disposal Solution for Control of a Giant Landslide and Dammed Lake in Yangtze River, China"

_water, doi:10.3390/w11091939_

Round 1
Reviewer 1 Report
â‘ Since this is case report, the structure is different from scientific article. Please reorganize this paper in form of report.
â‘¡ Further discussion of this case should be drew, especially about why these measures are effective in such a case, and the relationships or the differences between this case to others/other cases/other studies.
â‘¢ The conclusion should be added.
â‘£ Figure 6 is not clear in Time. And more details about the benefits (reduced loss) of theses measures should be given.
⑤ In line 169-173. There is a little confusion that flow peak discharge is increasing while it is decreasing in Liyuan Reservoir.
Author Response
Dear Editor and Reviewer #1
Please find the attachment on the Revision and the revised manuscript.
With best regards,
Yanlai Zhou

Reviewer 2 Report
The manuscript constitute the important issue of emergency disposal solution for control a giant landslide and dammed lake. It concerns the case report of the Yangtze River, China. Also as to show the importance of the presented issue, Authors send the supplementary file in the form of video. Manuscript is well prepared and deals with interesting and important issue, but it should include the following remarks. Line 122. In the text of the manuscript Mr. Ma Jianhua is mentioned several times, is he also the Author of the manuscript? The emergency disposal plan could be put in the reference section also. Line 197. How the presented approach will contribute to landslide management in other countries of the world? It should be underlined in the last paragraph of the manuscript.
Author Response
Dear Editor and Reviewer #2
Please find the attachment on the Revision and the revised manuscript.
With best regards,
Yanlai Zhou
